# A Preclinical Embryonic Zebrafish Xenograft Model to Investigate CAR T Cells in Vivo

**DOI:** 10.3390/cancers12030567

**Published:** 2020-02-29

**Authors:** Susana Pascoal, Benjamin Salzer, Eva Scheuringer, Andrea Wenninger-Weinzierl, Caterina Sturtzel, Wolfgang Holter, Sabine Taschner-Mandl, Manfred Lehner, Martin Distel

**Affiliations:** 1Innovative Cancer Models, St. Anna Children’s Cancer Research Institute, Zimmermannplatz 10, 1090 Vienna, Austria; susana.pascoal@ccri.at (S.P.); eva.scheuringer@ccri.at (E.S.); andrea.wenninger-weinzierl@ccri.at (A.W.-W.); caterina.sturtzel@ccri.at (C.S.); 2Christian Doppler Laboratory for Next Generation CAR T Cells, Zimmermannplatz 10, 1090 Vienna, Austria; benjamin.salzer@ccri.at (B.S.); manfred.lehner@ccri.at (M.L.); 3Development of Cellular Therapeutics, St. Anna Children’s Cancer Research Institute, Zimmermannplatz 10, 1090 Vienna, Austria; wolfgang.holter@ccri.at; 4Tumor Biology, St. Anna Children’s Cancer Research Institute, Zimmermannplatz 10, 1090 Vienna, Austria; sabine.taschner@ccri.at

**Keywords:** zebrafish xenografts, CAR T cells, CD19 CAR, in vivo imaging

## Abstract

Chimeric antigen receptor (CAR) T cells have proven to be a powerful cellular therapy for B cell malignancies. Massive efforts are now being undertaken to reproduce the high efficacy of CAR T cells in the treatment of other malignancies. Here, predictive preclinical model systems are important, and the current gold standard for preclinical evaluation of CAR T cells are mouse xenografts. However, mouse xenograft assays are expensive and slow. Therefore, an additional vertebrate in vivo assay would be beneficial to bridge the gap from in vitro to mouse xenografts. Here, we present a novel assay based on embryonic zebrafish xenografts to investigate CAR T cell-mediated killing of human cancer cells. Using a CD19-specific CAR and Nalm-6 leukemia cells, we show that live observation of killing of Nalm-6 cells by CAR T cells is possible in zebrafish embryos. Furthermore, we applied Fiji macros enabling automated quantification of Nalm-6 cells and CAR T cells over time. In conclusion, we provide a proof-of-principle study that embryonic zebrafish xenografts can be used to investigate CAR T cell-mediated killing of tumor cells. This assay is cost-effective, fast, and offers live imaging possibilities to directly investigate CAR T cell migration, engagement, and killing of effector cells.

## 1. Introduction

Infusion of T cells redirected to tumor cells by the expression of chimeric antigen receptors (CARs) is a promising and rapidly evolving strategy in cancer immunotherapy [1]. CARs are chimeric proteins comprising an extracellular antigen-binding domain, typically derived from an antibody, and intracellular signaling domains for activation of the T cells upon binding to the cognate antigen. The high potential of CAR T cells has been demonstrated in a series of clinical trials with B cell malignancies, most impressively with relapsed and refractory B cell acute lymphoblastic leukemia (B-ALL), with complete responses in more than 80% of the patients and durable responses in about half of the responders [2,3]. These groundbreaking successes with CAR T cells directed to the B cell-specific antigen CD19 have attracted huge interest within both academia and the pharmaceutical industry, and in 2017, first approval of CAR T cell therapy was obtained by the U.S. Food and Drug Administration (FDA) for therapy of pediatric ALL and non-Hodgkin lymphoma (NHL). However, the further development of CAR T cell therapy faces considerable hurdles. In the treatment of solid tumors, the efficacy of CAR T cells is unfortunately significantly lower, and the clinical translation of more effective CAR strategies is considerably delayed by the low tumor specificity and associated strong toxicities [4]. The preclinical development of strategies to improve the efficacy and safety of CAR T cells is currently typically performed in mice [5,6,7,8,9]. However, testing of CAR strategies in mice is expensive, laborious, and slow, and additional preclinical models would be beneficial to bridge the gap from in vitro studies to mouse xenografts.

Here, we explore the use of the zebrafish xenograft model for investigation of CAR T cell-mediated killing in vivo. Several cancer cell lines, including pancreatic cancer, ovarian carcinoma, glioma, breast cancer, prostate cancer, Ewing sarcoma, and leukemia, have been successfully xenotransplanted into zebrafish embryos and some of these xenograft models have subsequently been used to screen small compounds for anti-tumor effects [10,11,12,13,14,15,16,17]. Due to the absence of a functional adaptive immune system at embryonic and early larval stages of zebrafish development, transplanted cells are not rejected [18,19]. More recently, patient-derived xenograft models have been reported in zebrafish (zPDX) [20,21]. In one study, patient-derived colorectal cancer zebrafish xenografts were treated with the same chemotherapy as the corresponding patients and the authors speculate that zPDXs can be used to predict therapy outcome and to identify the best available therapy for the respective patient [21]. In contrast to small compounds, it is more complicated to investigate biologics or cellular therapies in zebrafish as they do not easily enter the organism upon administration into the water, but need to be injected. To our knowledge, CAR T cells have not been studied in combination with tumor cells in xenografted embryonic zebrafish. Here, we demonstrate that CAR T cell-mediated elimination of target cells can be monitored in living zebrafish embryos. We show that CD19-specific CAR T cells are able to kill pre B-leukemia cells (Nalm-6) in the zebrafish embryo and we developed a Fiji-based macro to quantify tumor cells and CAR T cells over time. Our results suggest that this novel zebrafish assay will be beneficial for cost-effective and fast preclinical evaluation of CAR T cells.

## 2. Results

### 2.1. Persistence of Nalm-6 Leukemia Cells in Zebrafish and CAR T Cell Killing Efficacy at 35 °C

To provide a proof-of-principle that CAR T cell-mediated killing can be observed in zebrafish embryos within a 24 hour assay, which is a typical time-span for in vitro tests, we decided to use a well-established pair in the CAR T cell field, a CD19-specific CAR already applied in the clinics for treatment of B cell malignancies and the CD19 expressing pre-B leukemia cell line Nalm-6 as target tumor cells [22].

To be able to monitor CAR T cells upon transplantation in zebrafish embryos, we labeled CAR T cells with DiI, a dye often used for human tumor cell transplants into zebrafish [23]. As this lipophilic dye stains cell membranes and could potentially interfere with CAR function, we first investigated whether DiI staining affects killing efficacy in vitro. Comparing DiI-labeled with non-labeled CAR T cells revealed a slight, but non-significant, reduction in killing efficacy at effector/target ratios between 0.5 and 8, indicating that DiI is a suitable label for our assay, as shown in Figure 1A–D.

As subsequent zebrafish xenografts will be carried out at 35 °C, we also investigated if killing of Nalm-6 cells by CD19 CAR T cells is altered at 35 °C compared to 37 °C in vitro. Using a 1:1 effector/target ratio, we observed a slight increase in killing efficacy at 35 °C compared to 37 °C within 24 hours, revealing that the lower temperature does not negatively affect the CD19 CAR T cell-mediated elimination of Nalm-6 cells (*n* = 3), as shown in Figure 1E. 

We next confirmed that our GFP-expressing Nalm-6 target cells persist upon xenotransplantation into zebrafish embryos for the duration of our 24 hour assay. For this, we injected approximately 200 Nalm-6 cells per zebrafish embryo and recorded images of xenografted embryos at 2 and 24 hours post injection (hpi), as shown in Figure 2A,A’. To be able to quantify cells in injected zebrafish, we applied a Fiji macro based on fluorescence in the tail region, where Nalm-6 cells injected into circulation accumulate. Using this tool, quantification of 37 injected embryos (two experiments) showed that there is no significant change in Nalm-6 cells within 24 hours, as shown in Figure 3C. In addition, immunostaining revealed that 66.8% ± 19.1% of Nalm-6 cells are positive for the proliferation marker Ki67 (*n* = 6 embryos, two experiments), as shown in Figure 2B,B’, and only 2.0% ± 1.7% showed active Caspase 3, demarcating apoptotic cells (*n* = 22 embryos, two experiments), as shown in Figure 2C,C’. 

Taken together, we show that neither staining with DiI nor a lower temperature at 35 °C prevent CAR T cells from eliminating target cells and that Nalm-6 cells persist in zebrafish at 35 °C for 24 hours in the absence of CAR T cells, indicating that these experimental parameters used in our zebrafish assay are permissive for investigating CAR T cell efficacy.

### 2.2. Elimination of Nalm-6 Cells by CD19 CAR T Cells can be Observed in Zebrafish Xenografts 

We next investigated whether CD19 CAR T cells can detect and eliminate Nalm-6 cells in zebrafish embryos. For this, approximately 200 Nalm-6 cells were injected into circulation of zebrafish embryos around 48 hours post fertilization (hpf), and 2 hours later around 200 DiI-labeled CAR T cells were additionally injected in each embryo. Images of successfully injected single embryos as evidenced by the presence of Nalm-6 and CAR T cells were acquired within 2 hpi and at 24 hpi. After 24 hpi, a reduction of Nalm-6 cells was visible in embryos injected with CD19 CAR T cells, as shown in Figure 3. Fluorescence-based quantification of the tail region, where Nalm-6 and T cells accumulate and interact, revealed a significant mean decrease of around 70% for Nalm-6 cells, as shown in Figure 3B–D (*n* = 51 embryos, four experiments). 

To confirm that killing was mediated by the CD19-specific CAR, we performed the same experimental strategy using T cells without a CAR. Using these mock T cells, we did not detect a significant change in Nalm-6 cell numbers at 24 hpi compared to 2 hpi, suggesting that killing is dependent on the presence of the CD19 CAR, and CAR-independent cytotoxicity of the T cells is negligible in our assay, as shown in Figure 3A,C (*n* = 38 embryos, two experiments). 

Using DiI-labeled T cells also allowed us to quantify T cell persistence over time using the fluorescence-based Fiji macro, which revealed that CD19 CAR T cells, as well as T cells without a CAR, showed no significant change in numbers over 24 hours in this assay, as shown in Figure 3C. Immunostaining for active Caspase 3 performed on CD19 CAR T cell-injected embryos at 24 hpi showed 5.4% ± 8.9% positive cells (*n* = 8 embryos, two experiments), as shown in Appendix A.

These results provide the proof-of-principle that CAR T cell-mediated killing can be investigated in living zebrafish embryos.

### 2.3. Live Imaging of CAR T Cell—Mediated Elimination of Nalm-6 Cells

One advantage this assay offers is the possibility to monitor CAR T and target cells at cellular resolution over time in the living organism. To temporally resolve the elimination of Nalm-6 cells, we carried out time-lapse imaging of the tail of zebrafish embryos co-injected with Nalm-6 and CD19 CAR T cells from 4 hpi to 19 hpi (Movie 1) (*n* = 2). At 4 hpi, many Nalm-6 cells have settled in the caudal vein plexus and several CD19 CAR T cells are present in the tail region of this embryo. Two hours after the start of recording, a wave of CAR T cells entered the posterior region of the dorsal aorta and subsequently the caudal vein and the caudal vein plexus. We quantified the number of Nalm-6 cells and the number of T cells in the tail region based on the covered fluorescent area over time, which revealed that a steady decrease of Nalm-6 cells was initiated, once the wave of CD19 CAR T cells had entered the caudal vein plexus, as shown in Figure 4A,C,E. Taking a time-lapse movie of a Nalm-6 xenotransplanted zebrafish embryo injected with mock T cells showed no obvious change in Nalm-6 cells, as shown in Movie 2 and Figure 4B,D,E (*n* = 2). 

In conclusion, we show that the distribution and cytotoxicity of CAR T cells can be studied in the zebrafish embryonic xenograft model in real time over several hours.

## 3. Discussion

CAR T cell-mediated therapy is a promising strategy to combat cancer and has shown efficacy for B-ALL and NHL. However, similar positive results are still missing for solid tumors and the design of CARs is still maturing. This emphasizes the need for higher throughput-suited preclinical models to test the plethora of possible strategies for improving the function of CAR T cells in the future before undertaking experiments in mice. 

Here, we have explored the use of zebrafish embryos xenografted with human tumor cells as an alternative assay to investigate CAR T cells towards their capability to eliminate target cells.

Using the well-established CD19 CAR together with Nalm-6 cells, we have shown that CAR T cell-mediated killing can be investigated in the embryonic zebrafish xenograft model. CD19 CAR T cells were able to kill Nalm-6 leukemia cells in transplanted zebrafish. In preliminary experiments, we also observed CD19 CAR T cell-associated killing using Raji cells as target cells. This promises that our assay can be used for future preclinical evaluation of novel CAR designs. 

Compared to the current gold standard, CAR T cell investigation in mouse xenografts, the embryonic zebrafish assay offers several complementing features. (1) It is cost-efficient as the cost per zebrafish embryo is low. (2) It is a fast assay. Killing of Nalm-6 cells could be observed in our hands with the CD19 CAR within 24 hours. (3) Many xenograft experiments can be performed from one tumor sample as only few tumor cells are injected (typically between 50 and 300 cells) per embryo. (4) Live imaging of the interaction between CAR T cells and target cells, as well as of the recruitment, accumulation, and localization of the CAR T cells at the tumor site is possible. This will be especially important to understand why some CAR T cell therapies fail, as has been recently observed for solid tumors, and how the function of CAR T cells can be improved. (5) Automation and higher throughput are possible for image acquisition and image analysis. Due to the small size of zebrafish embryos, they can be investigated in 96-well format and high content imaging is possible. (6) Zebrafish embryos are well suited for compound testing and it will be possible to screen for small compounds, which are able to act synergistically with CAR T cells, increasing their killing efficacy in the embryonic zebrafish xenograft model. (7) Furthermore, new CAR designs containing a small compound-based ON or OFF switch can be tested *in vivo* in zebrafish as small compounds can enter the embryo. 

However, there are also obvious differences and current limitations of the embryonic zebrafish xenograft model compared to the mouse model. The number of cells injected per zebrafish embryo is typically around 50 to 300 cells for tumor cells and CAR T cells, respectively, whereas in mice the cell numbers are typically about 4 orders of magnitude higher. Orthotopic injection can be challenging at embryonic stages as organs are still small. The duration of the assay is short and thus no escape of tumor cells from CAR T cell-mediated killing can be studied. Zebrafish possess only an innate immune system at embryonic and early larval stages. In addition, conservation of cytokines and their ability to cross-react between zebrafish and human still needs further investigation. Currently, there is also no humanized zebrafish model available, which possesses human immune cells. Furthermore, we are currently performing our assay at 35 °C and the slightly lower temperature could possibly influence the behavior of tumor and CAR T cells, although we have not detected negative effects on killing efficacy. Recently, a protocol allowing for the maintenance of adult zebrafish at 37 °C was published, promising that this will soon be possible for early developmental stages as well [24]. 

Taken together, we believe that our presented assay has the potential to bridge the gap between in vitro CAR T cell evaluation and mouse xenograft models and can be used as an in vivo filter to select for CAR designs, which should be tested more thoroughly in mouse. We also anticipate that ultimately, patient-derived xenografts in embryonic and larval zebrafish can be used as avatars to test CAR T cells, paving the road for future personalized CAR T cell therapy.

## 4. Materials and Methods 

### 4.1. In Vitro Methods/Preparation of Cells for Injection

Buffy coats from anonymous blood donations were purchased from the Austrian Red Cross, Vienna. Primary human T cells were isolated using the RosetteSep Human T Cell Enrichment Cocktail (STEMCELL Technologies, Vancouver, BC, Canada). Purified CD3^pos^ T cells were activated with Dynabeads Human T-Activator αCD3/αCD28 beads (Thermo Fisher Scientific, Waltham, MA, USA) at a bead-to-cell ratio of 1:1 according to the manufacturer’s instructions. Activated T cells were expanded in AIMV medium (Thermo Fisher Scientific, Waltham, MA, USA) supplemented with 2% Octaplas (Octapharma, Lachen, Switzerland), 1% L-Glutamine (Thermo Fisher Scientific, Waltham, MA, USA), 2.5% HEPES (Thermo Fisher Scientific, Waltham, MA, USA), and 200 IU/mL recombinant human IL-2 (Peprotech, Rocky Hill, NJ, USA), whereas half of the medium was exchanged every other day. Nalm-6 cells (kind gift from Dr. Sabine Strehl; St. Anna Children’s Cancer Research Institute (CCRI), Vienna, Austria) were maintained in RPMI-1640 GlutaMAX (Thermo Fisher Scientific, Waltham, MA, USA) supplemented with 10% fetal calf serum (FCS) (Merck, Darmstadt, Germany) and 1% penicillin-streptomycin (Thermo Fisher Scientific, Waltham, MA, USA). Lenti-X 293T cells (Takara, Kusatsu, Japan) were maintained in Dulbecco’s Modified Eagle Medium (DMEM) (Thermo Fisher Scientific, Waltham, MA, USA) supplemented with 10% FCS. Cell lines were regularly tested for mycoplasma contamination using the MycoAlert PLUS Mycoplasma Detection Kit (Lonza, Basel, Switzerland). Cell line authentication was performed by single nucleotide polymorphism (SNP)-profiling at Multiplexion GmbH, Heidelberg, Germany.

### 4.2. Construction of Transgenes

The CD19-specific CAR (FMC63.4-1BB.ζ) consists of the murine scFv FMC63, the hinge and transmembrane domain of CD8α (UniProt P01732 aa 138–206), the 4-1BB endodomain (UniProt Q07011 aa 214–255), and the CD3ζ signaling domain (UniProt P20963-3 aa 52–164), which included an additional Q65K mutation [25]. Synthetic DNAs were synthesized by GeneArt (Thermo Fisher Scientific, Waltham, MA, USA). Assembly of DNA molecules into complete constructs was performed using the NEBuilder® HiFi DNA Assembly Master Mix (New England BioLabs, Ipswich, MA, USA) according to the manufacturer’s instructions.

### 4.3. DiI Labeling of CAR T Cells

Primary human T cells or—in case of CAR T cell experiments—CD19-specific CAR T cells were labeled with the lipophilic dye DiI (CellTracker^TM^ CM-DiI cell labeling solution; Thermo Fisher Scientific, Waltham, MA, USA). Hereto, 5 to 15 × 10^6^ cells were spun down and washed twice with phosphate buffered saline (PBS) (Thermo Fisher Scientific, Waltham, MA, USA). The cell pellet was resuspended in the staining solution (4 µL CellTracker^TM^ CM-DiI in 1 mL PBS) and incubated for 10 minutes at 37 °C, followed by 15 minutes on ice and in the dark. The reaction was stopped with AIMV medium, the cells were spun down, and the supernatant was discarded. The cell pellet was resuspended in AIMV medium supplemented with 2% Octaplas, 1% L-Glutamine, and 2.5% HEPES. Thereafter, the cell suspension was filtered through a 35 µm cell strainer (BD Biosciences, Franklin Lakes, NJ, USA) and treated with 50 µg/mL DNase I (Roche, Basel, Switzerland) for 15 minutes at room temperature. Subsequently, T cells were spun down, washed twice with PBS supplemented with 2% FCS, and once with PBS. For subsequent xenotransplantation experiments, cells were set to a cell concentration of 200 cells/nL in PBS.

### 4.4. Lentivirus Production and Transduction

For production of VSV-G pseudotyped lentivirus, Lenti-X 293T cells were co-transfected with a puromycin-selectable pCDH expression vector (System Biosciences, Palo Alto, CA, USA), encoding either the second-generation αCD19 CAR (FMC63.4-1BB.ζ) or a reporter construct (firefly luciferase and GFP) and viral packaging plasmids, pMD2.G and psPAX2 (Addgene plasmids #12259 and #12260, respectively; gifts from Didier Trono), using the PureFection Transfection Reagent (System Biosciences, Palo Alto, CA, USA) according to the manufacturer’s instructions. Viral supernatants were collected approximately 36 hours and 50 hours after transfection and were concentrated 100-fold using the Lenti-X Concentrator (Takara, Kusatsu, Japan) according to the manufacturer’s instructions and frozen at −80 °C until further use. Primary human T cells were activated with αCD3/αCD28 beads 24 hours prior to lentiviral transduction. Cell culture plates were coated with RetroNectin (Takara, Kusatsu, Japan) according to the manufacturer’s instructions. Human T cells were set to a concentration of 0.5 × 10^6^ cells/mL and were exposed to viral particles (final dilution of viral suspension 1:2) for three days, followed by treatment with 1 µg/mL puromycin (Merck, Darmstadt, Germany) for two days to ensure high expression of the FMC63.4-1BB.ζ transgene. Firefly luciferase^pos^/GFP^pos^ Nalm-6 cells were generated by transducing the Nalm-6 cells with the respective lentiviral particles.

### 4.5. Cytotoxicity Assay

Cytolytic activity of primary human T cells was assayed with either a luciferase- or counting beads-based assay. 

Luciferase-based assay: 10,000 target cells (i.e., Nalm-6 cell line stably expressing firefly luciferase) were co-cultured for 4 hours at 37 °C with varying amounts of primary human T cells (E/T ratio as indicated) in white round-bottom 96-well plates in 100 µL RPMI-1640 (RPMI) medium without phenol red (Thermo Fisher Scientific, Waltham, MA, USA) supplemented with 10% FCS and 1% penicillin-streptomycin. Target cells without added effector cells served as a negative control (“targets only”). After co-culture, target cell viability was quantified by determining the residual luciferase activity. Hereto, culture plates were equilibrated for 10 minutes at room temperature. D-luciferin (Perkin Elmer, Waltham, MA, USA) was added at a concentration of 150 µg/mL. Bioluminescence was quantified after 20 minutes of incubation at room temperature on an ENSPIRE Multimode plate reader (Perkin Elmer, Waltham, MA, USA). Specific lysis was calculated with the following formula:
$$\%\mathrm{specific}\mathrm{lysis}=100-\left(\left(\frac{\mathrm{RLU}\;\mathrm{target}+\mathrm{effector}\;\mathrm{cells}}{\mathrm{RLU}\;\mathrm{target}\;\mathrm{cells}\;\mathrm{only}}\right)*100\right)$$
Counting beads-based cytotoxicity assay: 10,000 target cells (i.e., GFP-expressing Nalm-6) were co-cultured for 4 hours at 37 °C or 35 °C, as indicated, with 10,000 primary human T cells (= E/T ratio 1:1) in white round-bottom 96-well plates in 100 µL RPMIGlutaMAX supplemented with 10% FCS and 1% penicillin-streptomycin. Target cells not co-cultured with CAR-T cells served as a negative control (“targets only”). After co-culture, viable target cells were quantified by flow cytometry-based cell counting using counting beads. Hereto, 50 µL of staining solution (i.e., PBS, 0.2% human albumin (Merck, City, Germany), 20% (v/v) AccuCheck Counting Beads (Thermo Fisher Scientific, Waltham, MA, USA) and 30 µg/mL propidium iodide (PI) (Merck, Darmstadt, Germany)) was added to the sample and viable cells were quantified by flow cytometric analysis. For excluding dead cells, GFP+/PI- Nalm-6 cells were gated based on “targets only” control conditions. Specific lysis was calculated with the following formula:
$$\%\mathrm{specific}\mathrm{lysis}=100-\left(\left(\frac{\mathrm{cell}\;\mathrm{count}\;\mathrm{of}\;\mathrm{target}+\mathrm{effector}\;\mathrm{cells}}{\mathrm{cell}\;\mathrm{count}\;\mathrm{of}\;\mathrm{target}\;\mathrm{cells}\;\mathrm{only}}\;\right)*100\right)$$

### 4.6. Flow Cytometric Analysis

For the analysis of membrane-bound proteins, 1 × 10^5^ cells were resuspended in 50 µL fluorescence activated cell sorting (FACS) buffer composed of PBS supplemented with 0.2% human albumin (CSL Behring, King of Prussia, PA, USA) and 0.02% sodium azide (Merck, Darmstadt, Germany). After addition of the protein L-biotin conjugate (Genscript, Piscataway, NJ, USA), cells were incubated for 25 minutes at 4 °C and then washed twice with ice-cold FACS buffer. Cells were further incubated with streptavidin-PE (Thermo Fisher Scientific, Waltham, MA, USA) for 25 minutes at 4 °C and then again washed twice. Finally, cells were acquired on an LSR Fortessa instrument (BD Biosciences, Franklin Lakes, NJ, USA) and analyzed using FlowJo Software. Non-transfected cells served as negative controls.

### 4.7. Maintenance of Zebrafish

Zebrafish (*Danio rerio*) were maintained at standard conditions [26,27] according to the guidelines of the local authorities under licenses GZ:565304/2014/6 and GZ:534619/2014/4.

### 4.8. Xenotransplantation of Zebrafish

Mitfa^b692^/^b692^;ednrba^b140^/^b140^ embryos were raised at 28 °C until 48 hpf, dechorionated, anesthetized using 1× tricaine in E3 medium (0.16 g/L tricaine (Sigma-Aldrich Chemie GmbH, Munich, Germany), adjusted to pH 7 with 1M Tris pH 9.5, in E3) and embedded in 2% agarose (Biozym LE Agarose, Vienna, Austria) in E3 medium on a petri dish lid for xenotransplantation.

Xenotransplantation was performed using injection capillaries (glass capillaries GB100T-8P, without filament, Science Products GmbH, Hofheim am Taunus, Germany), pulled with a needle puller (P-97, Sutter Instruments, Novato, CA, USA), mounted onto a micromanipulator (M3301R, World Precision Instruments Inc., Friedberg, Germany), and connected to a microinjector (FemtoJet 4i, Eppendorf, Hamburg, Germany). 

GFP-expressing Nalm-6 cells and/or CAR T cells labeled with DiI were injected into circulation of zebrafish embryos at around 48 hpf and 50 hpf, respectively. 

After injection, xenografted embryos were selected for successful injection of labeled cells into circulation and kept at 35 °C for 24 hours. 

### 4.9. Imaging

Xenografted zebrafish were anesthetized in 1× tricaine/E3, placed always on the same side, and pictures were taken using an Axio Zoom.V16 fluorescence stereo zoom microscope with an Axiocam 503 color camera from Zeiss (Zeiss, Jena, Germany). Images were collected using the Zeiss image software ZEN, with a zoom factor of 3.3 at an exposure time of 32 ms. Images were acquired at 2 hpi and 24 hpi using the same settings.

### 4.10. Confocal Microscopy

Xenografted zebrafish were anesthetized in 1× tricaine/E3 and embedded in 1.2% ultra-low gelling agarose (Sigma-Aldrich Chemie GmbH, Munich, Germany) in a glass bottom imaging dish (D35-14-1.5-NJ, Cellvis, Mountain View, CA, USA) as described previously [28]. Images and time-lapse movies were recorded on an inverted Leica SP8 X WLL confocal microscope (Leica, Wetzlar, Germany) system using a 10× air objective.

### 4.11. Quantification of Cells Using a Fiji Macro

Cells were quantified using the open-source software ImageJ (Fiji package) according to the covered fluorescent area. Images from the Axio Zoom.V16 were saved in folders and imported to ImageJ as stacks. Regions of interest (ROIs) were marked by manually drawing lines with a width of 150 pixels which covered the whole tail posterior to the cloaca. With a macro, the ROIs were automatically converted to areas and saved to a directory for future reference. With another macro, the images were then automatically converted to grayscale, restricted with a lower threshold of 34, and the intensity measured within the restricted areas of the ROIs. The scripts for the macros are available in the online Appendix A. The individual data set of the replicates were joined and pre-processed with Python and further analyzed with Prism (Graphpad Software Prism 8, La Jolla, CA, USA).

### 4.12. Statistical Testing

To determine CAR T cell-mediated killing of target cells, statistical testing was done using Prism (Graphpad, Software Prism 8, La Jolla, CA, USA). The analysis for Figure 3 was carried out using a two-way ANOVA, Bonferroni post-hoc test. *p*-values < 0.05 were considered significant.

### 4.13. Image and Movie Rendering

Images were rendered using Photoshop CS6 (Adobe), Zeiss ZEN software, Leica LAS X software, Quicktime Pro, and Fiji.

### 4.14. Immunofluorescence

Xenografted embryos were initially fixed in 4% paraformaldehyde (PFA)/PBS (Electron Microscopy Sciences, Hatfield, PA, USA) overnight at 4 °C, then dehydrated in 100% methanol and stored at −20 °C. For immunostaining, zebrafish samples were gradually rehydrated to PBS through a methanol series (75% methanol/25% PBS, 50%/50%, 25%/75%, and 100% PBS), washed 2 × 5 minutes in PBSX (250 µL Triton X-100 in 50 mL PBS), washed once in distilled water, incubated for 7 minutes at −20 °C acetone, and for 1 hour in PBS/bovine serum albumin (BSA)/DMSO/Triton-X (PBDX). Subsequently, they were incubated overnight (O/N) at 4 °C with primary antibodies diluted in PBDX plus goat serum (GS), Ki67 (Ki-67 (8D5) mouse mAb #9449 Cell signaling Technology, Danvers, MA, USA) 1:400; Caspase 3 Antibody 1:100 (cleaved Caspase-3 (D175), Cell Signaling Technology, Danvers, MA, USA). Samples were washed 4 × 30 minutes in PBDX, incubated for 1 hour at room temperature (RT) with a secondary Alexa568 antibody (Alexa 568 anti-mouse: A-11019; Alexa 568 anti-rabbit: A-21069 Invitrogen, Carlsbad, CA, USA), washed 2 × 10 minutes with PBS, fixed for 5 minutes in 4% PFA, and washed 3 × 5 minutes in PBS. Finally, samples were mounted in Dako Fluorescence Mounting Medium (Dako, Agilent, Santa Clara, CA, USA) and imaged on a Leica SP8 confocal microscope (Leica, Wetzlar, Germany). 

## 5. Conclusions

Here, we explored the use of an embryonic zebrafish xenograft model to evaluate the efficacy of CAR T cell-mediated killing of target cells in vivo. In a proof-of-principle study, we were able to show that elimination of target cells by CAR T cells can be observed and quantified in living zebrafish embryos. This live imaging capability enables one to investigate CAR T cell distribution and localization, as well as interactions with tumor cells, and will give important insight at which step CAR T cell therapy might fail and needs to be improved. Furthermore, with easy compound uptake in zebrafish embryos, our assay is ideally suited to test novel CAR designs, relying on a small compound-mediated activation or deactivation. Taken together, we believe that our assay will complement existing approaches and bridge the gap between in vitro and mouse studies to evaluate novel CAR designs. 

## Figures and Tables

**Figure 1 cancers-12-00567-f001:**
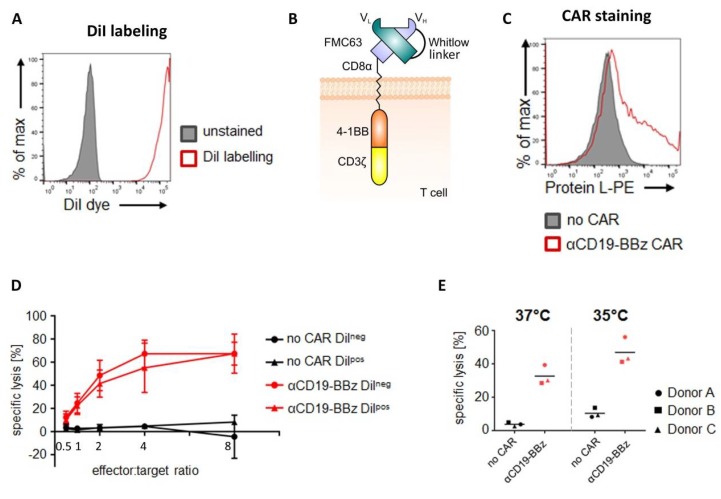
Chimeric antigen receptor (CAR) T cell mediated killing of Nalm-6 cells *in vitro*. (**A**) Efficacy of DiI-labeling in primary human T cells was analyzed by flow cytometry (excitation at 561 nm, detection with a 582/15 bandpass filter). Unlabeled T cells (“unstained”; filled grey histogram) served as a control. One representative experiment is shown. (**B**) Schematic of the CD19-specific second-generation CAR (FMC63.4-1BB.ζ) in which the FMC63-based scFv was fused to the hinge and transmembrane region of CD8α and the cytoplasmic domains of 4-1BB and CD3ζ. (**C**) Expression of the CD19-specific CAR in primary human T cells. Flow cytometric analysis using Protein L as a detection reagent. Mock T cells (“no CAR”; filled grey histogram) served as a negative control. One representative experiment of three independent experiments is shown. (**D**) Influence of DiI-labeling on the cytolytic activity of CAR T cells. Primary human T cells expressing either the CD19-specific CAR or no CAR and labeled or not labeled with DiI were co-cultured with CD19^pos^ Nalm-6 cells. Experiments were performed with E/T (effector/target) ratios of 0.5, 1, 2, 4, and 8, as indicated. Cytolytic activity was quantified using a luciferase-based cytotoxicity assay. Data are expressed as means ± SD of one experiment with three donors. (**E**) Influence of the temperature on the cytolytic activity of CAR T cells. Primary human T cells expressing either the CD19-specific CAR or no CAR were co-cultured at 35 °C or 37 °C with GFP^pos^/CD19^pos^ Nalm-6 cells at an E/T ratio of 1:1. Cytolytic activity was quantified by using a counting bead-based cytotoxicity assay. Data are expressed as individual data points (and their means) obtained from one experiment with three different T cell donors.

**Figure 2 cancers-12-00567-f002:**
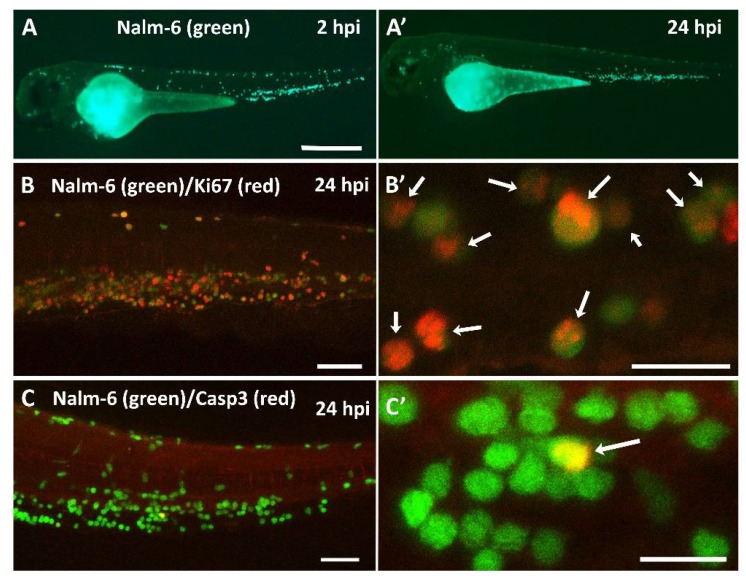
Nalm-6 xenografts in zebrafish embryos. GFP-expressing Nalm-6 cells (green) were injected into zebrafish embryos around 48 hours post fertilization (hpf). (**A**) An image was recorded at approximately 2 hours post injection (hpi) and again at 24 hpi (**A’**). (**B**) Immunostaining of the tail region using an anti Ki67 antibody (red) at 24 hpi, (**B’**) magnification of a region in B showing Ki67 positive Nalm-6 cells (arrows). (**C**) Immunostaining for apoptotic cells using an antibody against active Caspase 3 (red), (**C’**) magnification of a region in C. The arrow indicates a cell with active Caspase 3. Images in (**A**) were recorded on a Zeiss Axio Zoom.V16 fluorescence stereo zoom microscope, and in (**B**) and (**C**) on a confocal Leica SP8 WLL microscope. Images were rendered with Adobe Photoshop CS6. The scale bar in (**A**) represents 500 µm, in (**B** and **C**) 75 µm, and in (**B’** and **C’**) 25 µm.

**Figure 3 cancers-12-00567-f003:**
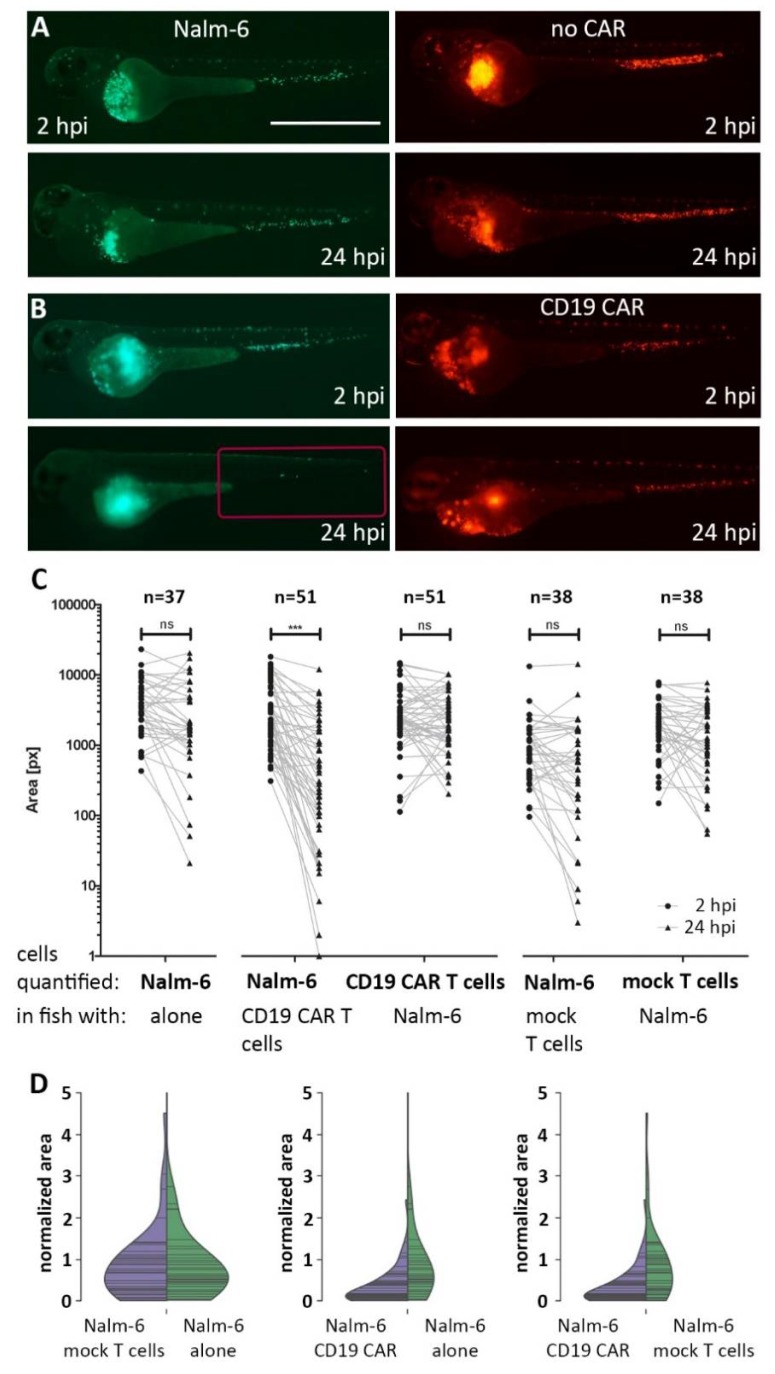
CAR T cell-mediated killing of Nalm-6 cells in zebrafish. Zebrafish embryos were injected with Nalm-6 cells (green) at approximately 48 hpf. Around 2 hours later, either mock T cells (without a CAR) (red cells in (**A**)) or CD19 CAR T cells (red cells in (**B**)) were injected and images were recorded within 2 hours and again at 24 hours post injection of T cells. (**C**) The numbers of Nalm-6 cells and T cells were quantified at 2 hpi and 24 hpi based on the fluorescent area covered by cells in the tail (red box in B) using Fiji. The two time points are connected by a line for each embryo. From left to right: Nalm-6 cells in zebrafish without any T cells; Nalm-6 cells (in zebrafish with CD19 CAR T cells), CD19 CAR T cells (in zebrafish with Nalm-6); Nalm-6 cells (in zebrafish with mock T cells), mock T cells (in zebrafish with Nalm-6) at 2 hpi and 24 hpi. (**D**) Violin plots normalized to the area covered by fluorescent cells at 2 hpi revealing the change in distribution at 24 hpi. Nalm-6 cells together with mock T cells or alone without T cells show similar distributions at 24 hpi, whereas Nalm-6 injected with CD19 CAR T cells show a reduction compared to Nalm-6 alone or Nalm-6 with mock T cells. Images were recorded on a Zeiss Axio Zoom.V16 fluorescence stereo zoom microscope and rendered with Adobe Photoshop CS6. Scale bar in (**A**) represents 1 mm.

**Figure 4 cancers-12-00567-f004:**
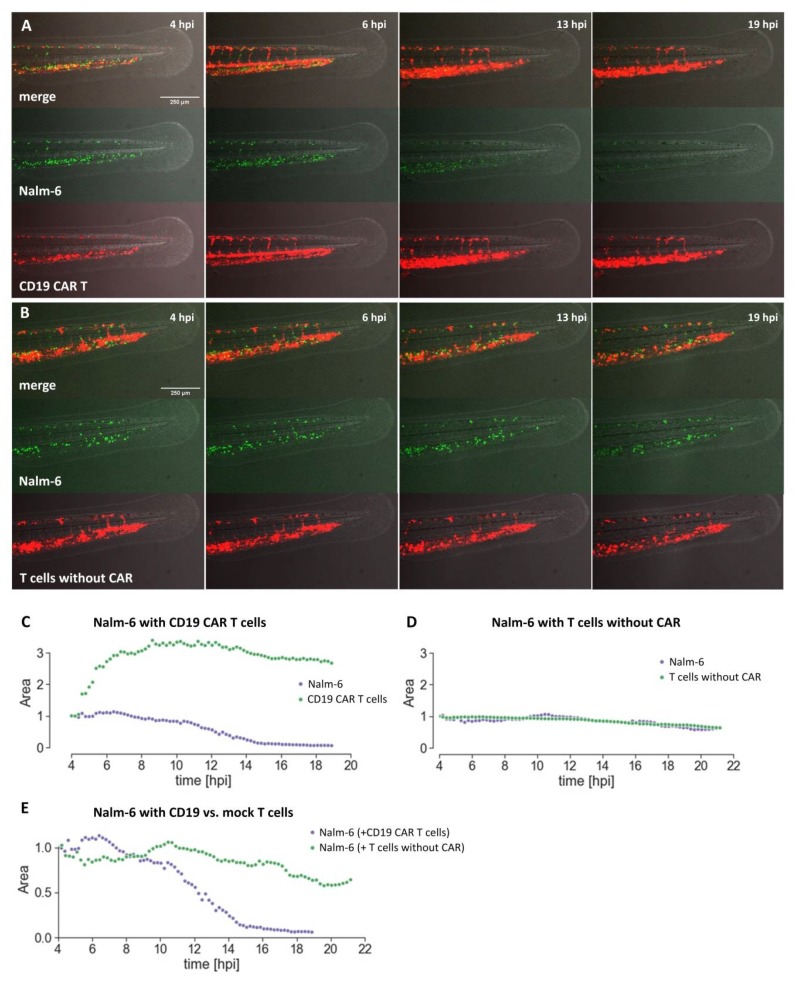
Quantifying specific elimination of Nalm-6 cells over time. Zebrafish embryos were injected with Nalm-6 cells (green) at approximately 48 hpf. Around 2 hours later, either CD19 CAR T cells (red cells in (**A**)) or T cells without a CAR (red cells in (**B**)) were injected and a time-lapse movie was recorded, starting approximately 4 hours post injection of T cells. Several time points of one embryo co-injected with CD19 CAR T cells (**A**) or mock T cells (**B**) are shown as single frames. (**C**) Quantification of cells over time based on the fluorescent area using Fiji. Nalm-6 (blue dots) in the presence of CD19 CAR T cells (shown as green dots), or the presence of T cells without CAR (**D**). (**E**) Nalm-6 in the presence of CD19 CAR T cells (blue) and T cells without CAR (green) over time. Images were recorded on a Leica SP8 X WLL confocal microscope and rendered with Adobe Photoshop CS6. Scale bar in (**A**) represents 250 µm.

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
