# Peer review of "A Preclinical Embryonic Zebrafish Xenograft Model to Investigate CAR T Cells in Vivo"

_cancers, 2020, doi:10.3390/cancers12030567_

Round 1
Reviewer 1 Report
In this article, Pascoal and colleagues for the first time show that the zebrafish can be used as a preclinical model organism to investigate CAR T cells in vivo. They use the CD19 expressing pre-B leukemia cell line Nalm-6 as target tumor cell line and CD19-specific CAR T, which they co-inject into zebrafish embryos to analyze specific lysis of Nalm-6 cells. This is an elegant study which however could be improved with some further experiments. Most importantly, I feel that a second cell line or even primary patient material if available should be injected to finally proof the usability of the zebrafish in this aspect since only one cell line has been used.
Additional comments:
The Berman group has already used the ZF as an in vivo model to study primary ALL samples. This may be mentioned in the introduction or at least referenced since this is more related to their work as the reference about patient-derived colorectal cancer cells. In regard to references, it might also be important to cite articles about CAR T testing in mice. Are the differences in Figure 1E between 37°C and 35°C significant? Would this indicate that lowering the body temperature might increase the efficacy in human CAR T cell treatment? Can the authors speculate? For Figure 2, quantification has apparently been performed, but no graphs/data points are displayed. The authors might want to provide this and also indicate, how many times these experiments were repeated. What about the viability of CAR T cells in ZF? Could the authors perform co-staining with e.g. Caspase 3 as it has been shown in Figure 2 to proof this? It might be obvious from the data points regarding the number of cells at 24 hpi compared to 2hpi, but nevertheless it may be useful to provide data about cell viability after transplantation. Can the authors exclude that tissue macrophages are involved in any of the processes shown in this article? How may times were experiments in Figures 3 and 4 repeated? It would be extremely elegant if the authors could show in vivo evidence of the actual lysis process by e.g. performing live-cell imaging at higher magnifications, especially since the zebrafish easily allows these kinds of analyses.Author Response
Please see attachment

Reviewer 2 Report
Overview
The authors investigate how human cancer cell xenografts in zebrafish embryos can be used to evaluate chimeric antigen receptor (CAR) T cell-mediated killing using a using a CD19-specific CAR and Nalm-6 B cell leukemia cells (which express CD19). The authors demonstrate efficient killing by visualization of embryos and develop an ImageJ macro software tool to semi-automate quantification of images of cells in fish over time. The study is of interest both technically as well as conceptually for the evaluation of CAR systems in a medium-throughput and facile way using zebrafish. There are only a few minor concerns, which should be addressed.
Major comments
The authors should compare their in vivo quantification of labelled CAR-T cell and Nalm-6 cells by another method, e.g. ex vivo enumeration as in Corkery et al., 2011 Brit. J. Haematol. PMID: 21517816. The best way to do this would be to dissociate groups of 5-10 fish per tube/per time point, and then normalizing the data to the number of fish per tube dissociated.The ImageJ macro should be included as supplemental data (e.g. as a text file).
Minor comments
In the methods, the CAR-T labelling section in 4.3 forgets to mention that the primary cells for CAR-T experiments are transduced with the anti-CD19 CAR.Although potentially outwith of the current study – quantitative PCR for human T cell transcripts for cytokines involved in functional CAR T activity (e.g. like IL2) would be a nice addition to the manuscript
Reviewer 3 Report
The authors presented a novel assay based on 24 embryonic zebrafish xenografts to investigate CAR T cell-mediated killing of human cancer cells. Using CD19-specific CAR and Nalm-6 leukemia cells, they show that live observation of killing of Nalm-6 cells by CAR T cells is possible in zebrafish embryos.
In Fig1, the authors compared DiI-labeled and non-labeled CAR T cells in vitro and showed a slight, but non-significant reduction in killing efficacy at effector/target ratios between 0.5 and 8. Also, a reduction of temperature from 37oC to 35oC with E:T-ratio of 1:1 showed an increase in specific lysis of Nalm-6 cells in vitro. However, the authors did not comment on why the lower temperature resulted in both no-CAR and CD19 CAR-T increased lysis of Nalm-6 cell? Are they functioning abnormally or something unknown of the T cells were changed that affecting their T-cell killing properties, in both no-CAR and CD19-CAR?
In fig2, Ki67 mouse mAb and Caspase 3 antibody also detected cells that are not green, which were not Nalm-6 cells, but the authors did not address if those were fish cells or something else that detected by these antibodies? The authors should comment on the specificity of these assays and whether the conclusion was still valid about the proliferation and apoptotic status of the human Nalm-6 cells?
In fig3, the image of Nalm-6 injected xenograft (Fig3a) showed the distribution of Nalm-6 cells in no-CAR group were largely located in the yolk pericardial region and very little amount of Nalm-6 cells were distributed in the red box tail region in Fig3b (primarily the CHT region). This is also true for the CD19 CAR-T group (Fig6b). It is not convincing to just compare the Nalm-6 cells in the tail region because there were quite a lot of cells in the dorsal trunk as well as the pericardial regions and the head region as well. Based on the selected images, the tail region showed the obvious difference in term of the Nalm-6 cells, however, based on the quantitation data (Fig 3c), it was a wide range from very small vs very large value in cellular area in the tail region. The method the authors chose only to quantify the Nalm-6 cells in the tail region, however, was not fully justified. If the authors could include the other parts (i.e. head, trunk, yolk areas) of the zebrafish for quantification, it would provide more convincing evidences to support the conclusion that the numbers of Nalm-6 cells were decreased by CD19 CAR-T treatment.
Also, in the same figure 3c, the Nalm-6 cell areas at 2hpi (before the introduction of no-CAR or CD19-CAR T cells) is consider as the initial value of transplanted Nalm-6 cells so that it can be compared to the cell areas at 24hpi. The control group Nalm-6 cell (area) has an average of about 3,500 and the CD19 CAR-T group has an average of about 2,000, and the Mock T-cell group has average of about 900. The initial value of each group should have a similar range of Nalm-6 cells transplanted but the data did not indicate so. Also, the amount of Nalm-6 value at 24hpi for mock-T cell vs CD19 CAR-T group have no significant difference, if the variability of initial transplanted Nalm-6 cells were so large, it would be doubtful to draw any meaningful conclusion. Obviously the authors chose the most dramatic effect of the CD19 CAR-T group in the CHT region in Fig3b but the quantitative data in Fig3c showed there was hugh variability within the treatment group. If the variability of transplanted Nalm-6 cells were so large, then the conclusion for the effect of treatment group will be compromised as well as the conclusion. The authors should address that vigorously, otherwise, the results were not conclusive.
In addition, the authors claimed that they developed a software tool enabling automated quantification of Nalm-6 cells and CAR T cells over time, however, the tool was protocol using Fiji macro of ROI intensity measurement. It should not be over stated as development of a software tool instead.
In Fig4, the time lapse images showed reduction of Nalm-6 cells in the tail (largely the CHT) region of CD19 CAR-T xenograft. However, the authors did not describe the zebrafish sample number per treatment group in Fig4c,d,e. If the data for Fig4c,d,e were from 1 xenograft embryo, that would not be conclusive enough because Fig3c already showed that the xenografts were highly variable in Nalm-6 cells within different embryos. Without evaluation of multiple zebrafish xenografts per treatment group, the authors should not make any conclusion like that. Based on the variability we see in Fig3c, the sample number should be comparable to the n (i.e. 37 to 51 xenografts) in order to draw any conclusion in Fig4.
Suppl Video legend was confusing and need to be corrected in the description in Video S2. Was it not mock T cells (red) in the legend?
Round 2
Reviewer 1 Report
The authors answered all of my questions. However, I still feel that the second cell line they now tested should at least be mentioned to make their findings more sound.
